# Collection of biospecimens from the inspiration4 mission establishes the standards for the space omics and medical atlas (SOMA)

Eliah G. Overbey [1,2,3,4], Krista Ryon[1], JangKeun Kim [1,2], Braden T. Tierney [1,2], Remi Klotz [5,6], Veronica Ortiz[5], Sean Mullane[7], Julian C. Schmidt[8,9], Matthew MacKay[1], Namita Damle[1], Deena Najjar [1], Irina Matei [10,11], Laura Patras[10,12], J. Sebastian Garcia Medina[1], Ashley S. Kleinman[1], Jeremy Wain Hirschberg[1], Jacqueline Proszynski[1], S. Anand Narayanan[13], Caleb M. Schmidt[8,9,14], Evan E. Afshin[1], Lucinda Innes[1], Mateo Mejia Saldarriaga[15], Michael A. Schmidt[8,9], Richard D. Granstein[16], Bader Shirah [17], Min Yu[5,6], David Lyden [10,11], Jaime Mateus[7] & Christopher E. Mason [1,2,3,18,19] ✉

The SpaceX Inspiration4 mission provided a unique opportunity to study the impact of spaceflight on the human body. Biospecimen samples were collected from four crew members longitudinally before (Launch: L-92, L-44, L-3 days), during (Flight Day: FD1, FD2, FD3), and after (Return: R + 1, R + 45, R + 82, R + 194 days) spaceflight, spanning a total of 289 days across 2021-2022. The collection process included venous whole blood, capillary dried blood spot cards, saliva, urine, stool, body swabs, capsule swabs, SpaceX Dragon capsule HEPA filter, and skin biopsies. Venous whole blood was further processed to obtain aliquots of serum, plasma, extracellular vesicles and particles, and peripheral blood mononuclear cells. In total, 2,911 sample aliquots were shipped to our central lab at Weill Cornell Medicine for downstream assays and biobanking. This paper provides an overview of the extensive biospecimen collection and highlights their processing procedures and long-term biobanking techniques, facilitating future molecular tests and evaluations.As such, this study details a robust framework for obtaining and preserving high-quality human, microbial, and environmental samples for aerospace medicine in the Space Omics and Medical Atlas (SOMA) initiative, which can aid future human spaceflight and space biology experiments.

Our human space exploration efforts are at a unique transition point in history, with more crewed launches and human presence in space than ever before[1]. We can attribute this to the commercial spaceflight sector entering an industrial renaissance, with multiple companies forming collaborative and competitive networks to send commercial astronauts into space. The recent advancements in human space exploration offer a significant chance to gather more biological research samples and enhance our comprehension of spaceflight's effects on

human health. This is vital, as our knowledge of the biological reactions to the unique environment of space, marked by microgravity and the space radiation, remains incomplete[2]. The impact of spaceflight on human health includes musculoskeletal deconditioning[3], cardiovascular adaptations[4], vision changes[5], space motion sickness[6], neurovestibular changes[7], immune dysfunction[8], and increased risk of rare cancers[9], among other changes[2]. However, we are still at the initial stages of documenting biological responses to spaceflight exposure at the molecular and cellular resolution.

Prior work has characterized molecular changes that occur during spaceflight in astronauts. These include changes in cytokine profiles[8,10,11], urinary albumin abundance[12], and hemolysis[13]. Furthermore, multiomic assays have provided genomic maps of structural changes in DNA[14–16], RNA expression profiles[11,17,18], sample-wide protein measurements[17,19,20], and metabolomic status[17]. Additionally, International Space Station (ISS) surfaces have been studied with longitudinal microbial profiles to track microbial pathogenicity and evolution to assess their potential influence on crew health[21,22]. To better improve our understanding of both human and microbial biology in space, it is critical that these analyses continue and expand as more spacecraft and stations are built and flown.

Combining and comparing work from prior missions in these new spacecraft and stations is especially important to overcome the small sample sizes and highlights a need for standardization between missions. In addition, recruiting large cohorts of astronauts is difficult, as the ISS typically can only house up to six astronauts at a time. As of the time of writing, only 647 humans have been to space, starting with the launch of Yuri Gagarin in 1961. Studies have spanned the Vostok program, Project Mercury, the Voskhod program, Project Gemini, Project Apollo, the Soyuz program, the Salyut space stations, MIR, the Space Shuttle Program, SkyLab, Tiangong Space Station, and the ISS. From the breadth of experiments that have been performed on the ISS, only a minority have specifically been human research-oriented[23,24], and just a subset involve omics studies. The NASA Twin Study created the most in-depth multi-omic study of astronauts prior to Inspiration4, but was limited to one astronaut and one ground control[17]. These factors have limited the statistical power of astronaut omic experiments and increased the difficulty of providing robust scientific conclusions. Standardizing biospecimen collections across multiple missions will create larger sample sets needed to draw these conclusions.

Here, we establish the standard biospecimen sample collection and banking procedures for the Space Omics and Medical Atlas (SOMA). A key goal of SOMA is to standardize biospecimen collection and processing for spaceflight, to generate high-quality multi-omics data across spaceflight investigations, and to enable follow-up experiments with viably frozen cells and biobanked samples. This paper provides sample collection methods built for standardized field collections across different crews and missions. We cover the decentralized sample collection process from the Inspiration4 (I4) mission from the point of collection to the preservation of samples for downstream biochemical and omics processing at our centralized lab at Weill Cornell Medicine (WCM). These protocols can produce synchronized datasets with enhanced statistical strength, amplifying our scientific findings from spaceflight studies. Additionally, we showcase metrics related to sample collection outputs, references of past astronaut sample collections in scientific publications, and suggestions to refine sample collection for upcoming missions. In its inaugural use case, these samples were collected from the I4 astronaut cohort and are currently in use for several other missions (Polaris Dawn, Axiom-2), which will enable continued utilization for future crewed space missions.

## Results

### Biospecimen collection overview

We formulated and executed a sampling plan that spans a wide range of biospecimen samples: venous blood, capillary dried blood spots (DBSs), saliva, urine, stool, skin swabs, skin biopsies, and environmental swabs (Fig. 1a). The collection of various types of samples covered the scope of previous assays on astronaut samples (Supplementary Table 1), but also enabled newer omics technologies, such as spatially resolved, single-molecule, and single-cell assays.

For the I4 mission, sample collection spanned three-time points pre-launch (L-92, L-44, L-3 days), three-time points during flight (Flight Day 1 (FD1), FD2, FD3), and four-time points post-return (R + 1, R + 45, R + 82, R + 194 days). Venous blood, urine, stool, and skin biopsies were collected during ground timepoints only, while capillary DBSs, saliva, and skin swabs were collected both on the ground and during flight (Fig. 1b). Environmental swabs of the Dragon capsule were collected pre-flight in the crew training capsule and during flight in the spacecraft launched from Cape Canaveral (Fig. 1b).

Samples were collected across a variety of locations based on the crew's training and travel schedule. L-92 and L-44 were collected in Hawthorne, CA at SpaceX Headquarters, L-3 and R + 1 were collected at Cape Canaveral, FL at a facility near the launch site. FD1, FD2, and FD3 were collected inside the Dragon capsule while in orbit. R + 45 was collected at the crew members' individual locations (which spanned the US States NY, NJ, TN, and WA), R + 82 was collected at Weill Cornell Medicine, NY and R + 194 was collected at Baylor College of Medicine, TX (Fig. 1c). All samples from ground timepoints were processed within 16 hours of collection. All samples from the flight were processed immediately after retrieval from the flight.

Samples were stored at −80 °C at each site immediately after processing and aliquoting (processing steps are outlined below). All samples were shipped via FedEx using Overnight Shipping to Weill Cornell Medicine for storage in the Cornell Aerospace Medicine Biobank (CAMbank) within 1 week of collection on dry ice in styrofoam boxes and spent <2 days in transit. No samples were shipped on a Thursday or Friday to avoid weekend shipping delays. All samples arrived at Weill Cornell Medicine with dry-ice still in the packaging and samples frozen. Samples were immediately transferred to −80 °C for long-term storage and biobanking. Any exceptions to this process will be noted below. In total, we collected 2,911 sample aliquots, which were then processed in our central lab at Weill Cornell Medicine for downstream assays (Supplementary Table 2). Some samples were processed immediately at field laboratory facilities (Supplementary Data Fig. 1). DNA and RNA yields are also reported below from collection kits (saliva and stool) after arrival at WCM.

### Blood collection and derivatives

Blood was collected using a combination of venipuncture tubes to collect venous blood and contact-activated lancets to collect capillary blood from the fingertip. Each crew member provided blood samples, collected into one BD PAXgene blood RNA tube (bRNA), four BD Vacutainer K2 EDTA tubes, two BD Vacutainer cell preparation tubes (CPTs), one Streck cell-free DNA tube (cfDNA BCT), one BD Vacutainer serum separator tube (SST), and one DBS card per time point. Tubes were transported from the collection site to the field laboratory at room temperature. From these tubes, whole blood, plasma, PBMCs, serum, and cell pellet samples were collected (Supplementary Table 3). Sample yields are reported below. Samples were aliquoted for long-term storage and biobanking (Supplementary Table 4).

bRNA tubes were collected in order to isolate total RNA using the PAXgene blood RNA kit (Fig. 2a). Yield ranged from 3.04 to 14.04 μg/tube of total RNA across all samples, and the RNA integrity number (RIN) ranged from 3.2 to 8.5 (mean: 6.95) (Fig. 2b). RNA was stored at −80 °C after extraction. The collection of total RNA enables a variety of downstream RNA profiling methods. It will allow comparative studies to prior RNA-sequencing performed on astronauts, particularly snoRNA & lncRNA biomarkers analyzed from Space Shuttle era blood[25,26], mRNA & miRNA measured during the NASA Twin Study[17,18], and whole blood RNA arrays from the ISS[27,28]. Additionally, RNA yields

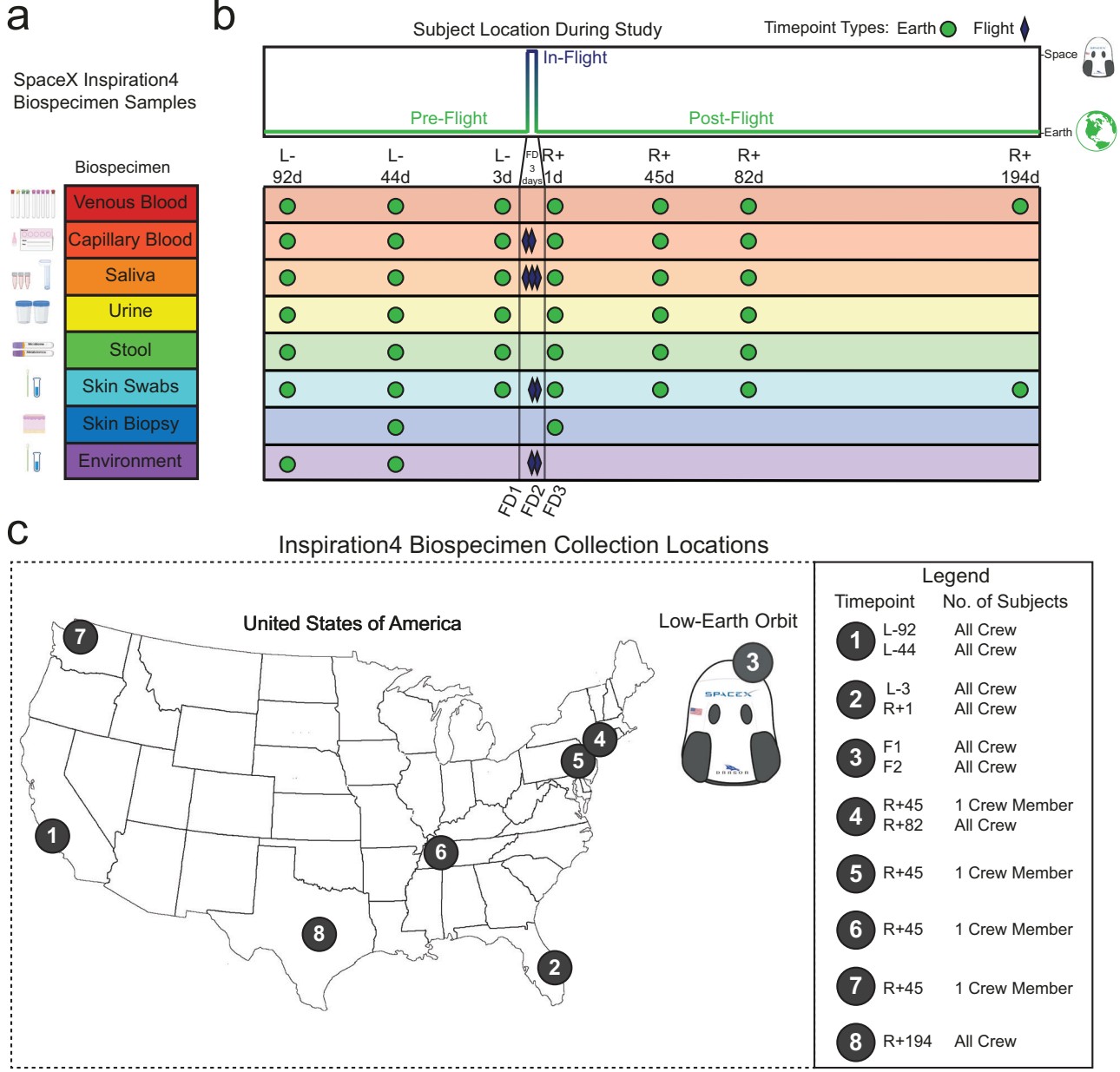

**Fig. 1 | Biospecimen Samples and Collection Locations. a** List of biospecimen samples collected over the course of the study. **b** Timepoints for each biospecimen sample collection. "L-" denotes the number of days prior to launch. "R + " denotes the number of days after return to Earth. "FD" denotes which day of the flight a sample was collected. **c** Location of each collection timepoint.

are more than sufficient to perform direct-RNA sequencing using Oxford Nanopore Technologies (ONT) platforms, which require 500 ng of total RNA per library (Manufacturer's protocol, ONT kit SQK-RNA002). This enables the study of RNA modification changes during spaceflight to create epitranscriptomic profiles for the first time in astronauts.

Four K2 EDTA tubes were drawn at each timepoint from each crew member (Fig. 2c). One K2 EDTA tube was submitted to Quest Diagnostics to perform a complete blood count (CBC, Quest Test Code: 6399). One tube was used to isolate extracellular vesicles and particles (EVPs) for proteomic quantification (Fig. 3a). Total EVP quantities varied from 2.71-28.27 ug (Fig. 2d). Two K2 EDTA tubes were used to isolate PBMCs for single-cell sequencing (10X Chromium Single Cell Multiome ATAC + Gene Expression and Chromium Single Cell Immune Profiling workflows). After collection, a Ficoll separation was performed to isolate PBMCs, which ranged from 340,000-975,000 cells

per mL of blood (Fig. 2e). One prior single-cell gene expression experiment, NASA Twin study, was performed on astronauts, which found immune cell population specific gene expression changes and a correlation with microRNA signatures[11,18].

Additional PBMCs, plasma, and serum were collected from CPTs (Fig. 4a), cfDNA BCTs (Fig. 4d), SSTs (Fig. 4c), as well as red blood cell pellets. CPTs were spun and aliquoted according to the manufacturer's instructions (Fig. 3b). Plasma volume per tube ranged from 3000-14,000 uL per tube (Fig. 4d). Because of technical issues in the sample processing procedure, three instances occurred where plasma retrieval from the CPT tubes was not possible. Plasma can be used to validate or refute previous studies, including cytokine panel[10,29], exosomal RNA-seq[25,26], extracellular vesicle microRNA[30], and proteomic[20,31–33] results. PBMCs were also collected, aliquoted into 6 cryovials per CPT, and stored in liquid nitrogen after slowly cooled in a Mr. Frosty to -80 °C. These can be used to follow-up on previous studies on adaptive

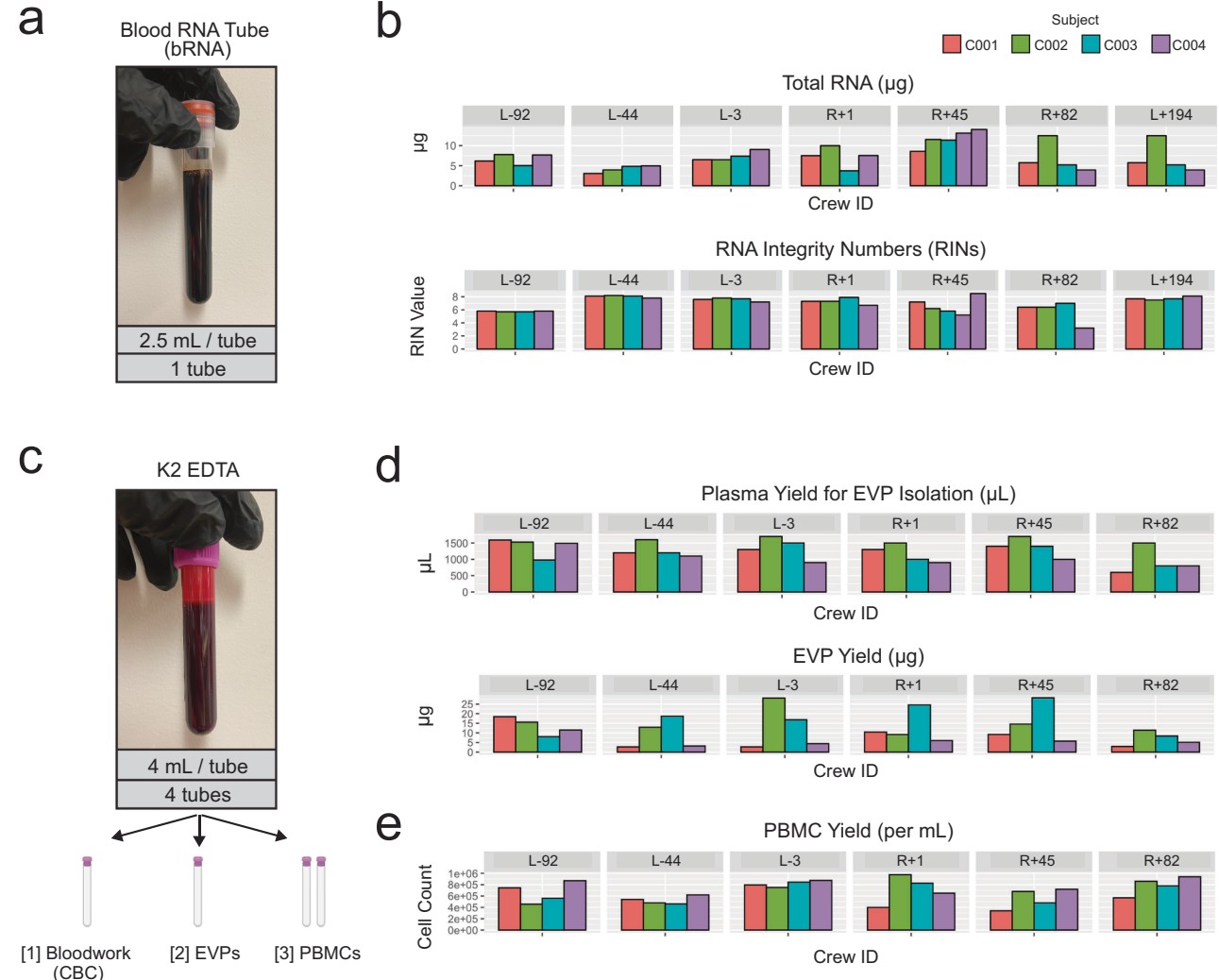

**Fig. 2 | bRNA and K2 EDTA Tubes. a** One 2.5 mL bRNA tube was collected per crew member at each ground timepoint. **b** bRNA tube total RNA yields per sample (µg) and RINs. **c** Four K2 EDTA tubes were collected per member at each ground timepoint. One tube was used for a CBC, one tube was used to isolate EVPs, and two tubes were used for isolation of PBMCs. **d** Plasma and EVP yields from the "[2] EVPS" tube on Fig. 2c. **e** PBMC yields per mL from the "[3] PBMCs" tubes on Fig. 2c.

immunity, cell function, and immune dysregulation[8,34–39]. The remaining red blood cell pellet mixtures from below the gel plug in each CPT Tube were stored at −20 °C.

cfDNA BCT (Streck) tubes were collected to isolate high-quality cfDNA from plasma. cfDNA BCTs were spun and aliquoted according to the manufacturer's instructions (Fig. 3c). The remaining cell pellet mixture was frozen at −20 °C. Plasma volume per timepoint ranged from 1500-5000 uL (Fig. 4e). 500 uL aliquots were frozen at -80 °C. cfDNA extracted from these tubes can be analyzed for fragment length, mitochondrial or nuclear origin, and cell type or tissue of origin[40–42].

The SST was spun and aliquoted according to the manufacturer's instructions (Fig. 3d). Serum volume ranged from 2000-8000 uL per timepoint (Fig. 4f). Similar to plasma, serum can be allocated for cytokine analysis and can also be used to perform comprehensive metabolic panels, including one we used at Quest (CMP, Quest Test Code: 10231) for metrics on alkaline phosphatase, calcium, glucose, potassium, and sodium, among other metabolic markers. The remaining cell pellet mixture from each SST tube was stored at −20 °C.

In addition to venous blood, capillary blood was collected onto a DBS card using a contact-activated lancet pressed against the fingertip (Fig. 5a). Capillary blood was collected onto a Whatman 903 Protein Saver DBS card to preserve nucleic acids and proteins. Each of the five spots on the DBS card hold 75-80uL of capillary blood, however, the amount of capillary blood collected across timepoints varied (Fig. 5b, c) according to how much blood could be collected before the lancet-puncture closed.

## Saliva collection

Saliva was collected at the L-92, L-44, L-3, FD1, FD2, FD3, R + 1, R + 45, and R + 82 timepoints using two methods. First, saliva was collected using the OMNIgene Oral Kit (OME-505), which preserves nucleic acids (Fig. 6a) during the ground timepoints. From these samples, DNA, RNA, and protein were extracted. DNA yield ranged from 28.1 to 3187.8 ng, RNA yield from 396.0 to 3544.2 ng (less the two samples had concentrations too low for measurement), and protein concentration from 92.97 to 93.15 ng.

Second, crude saliva (i.e., saliva with no preservative added) was collected into a 5 mL DNase/RNase-free screw top tube during the ground and flight timepoints. Saliva volume varied from 150 to 4000 uL per tube (Fig. 6b). Crude saliva was also collected during flight (FD2 and FD3), in addition to the ground timepoints.

Saliva collections have been conducted throughout spaceflight studies for assessing the immune state of crews, particularly in the context of viral reactivation. Previously identified viruses that reactivate during spaceflight include Epstein–Barr, varicella-zoster, and

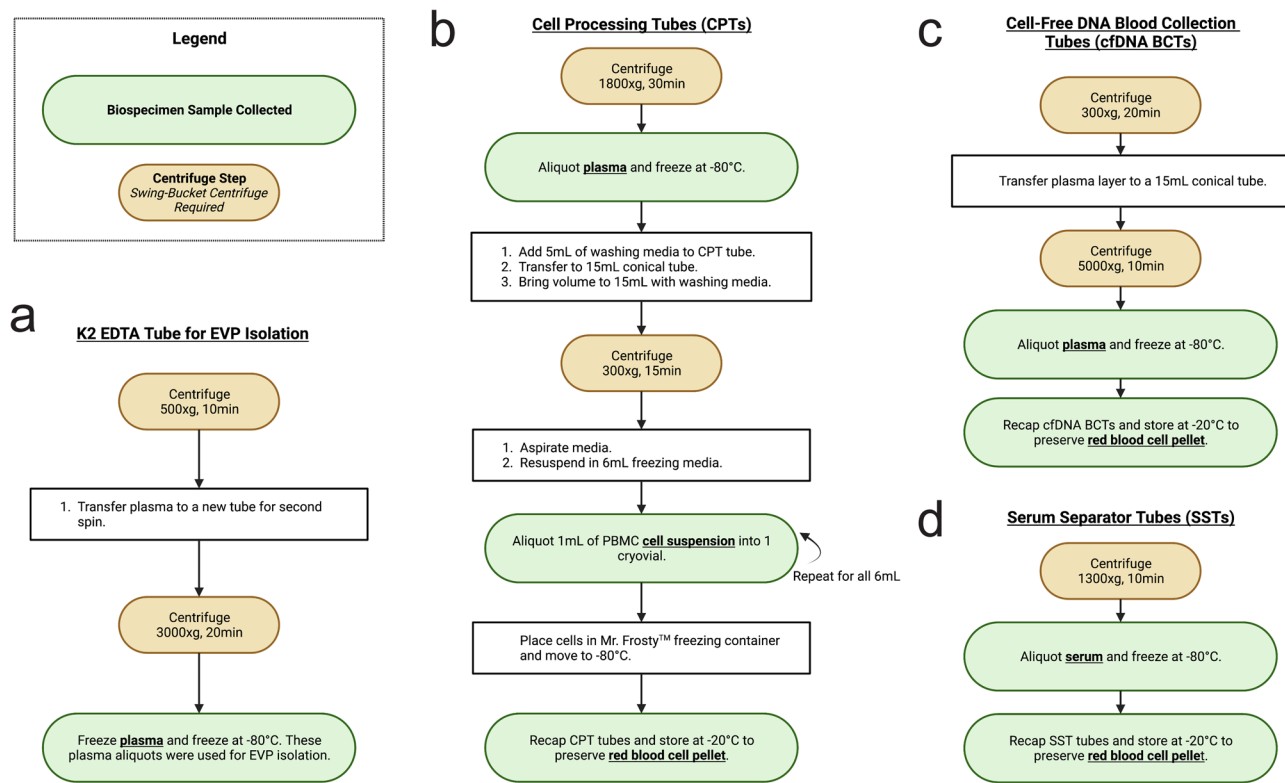

**Fig. 3 | Tube Processing Steps.** Centrifuge (brown circles) and aliquoting (white and green boxes and circles) protocols for **a** K2 EDTA tubes designated for EVP isolation **b** CPTs **c** cfDNA BCTs ans.d **d** SST.

cytomegalovirus[43]. Responses to reactivation of these viruses can be asymptomatic, debilitating, or even life-threatening, thus assessing these adaptations is beneficial in understanding viral spaceflight activity as well as crew health. In addition to viral nucleic acid quantification, numerous biochemical assays can also be performed, including measurements of C-reactive protein (CRP), cortisol, dehydroepiandrosterone (DHEA), and cytokines, among others[10,43–46].

### Urine collection
Urine was collected in sterile specimen cups at the L-92, L-44, L-3, R + 1, R + 45, and R + 82 timepoints. Specimen cups were collected 1-2 times per day. For preservation, urine was aliquoted and stored at -80 °C. Half the urine had Zymo Urine Conditioning Buffer (UCB) added before freezing, to preserve nucleic acids. Samples yielded 23–155.5 mL of crude urine and 21 - 112 mL of UCB urine per specimen cup (Fig. 7a). Urine was split into 1 mL - 15 mL aliquots before freezing at -80 °C.

A wide variety of assays can be performed on urine samples. Previous studies have included viral reactivation[43,46,47], urinary cortisol[48,49], iron and magnesium measurements[50–53], bone status[54–56], kidney stones[54,55], proteomics[11], telomere measurements[57], and various biomarkers and metabolites[17,49,58–61].

### Stool collection
Stool was collected at the L-92, L-44, R + 1, R + 45, and R + 82 timepoints. Stool samples were stored into two collection containers at each timepoint, one DNA Genotek OMNIgene Gut (OMR−200) kit with a preservative for metagenomics and another (ME−200) with a preservative for metabolomics (Fig. 7b). Stool was the least consistent sample collected due to the limited windows available for sampling during collection timeframes. DNA and RNA were extracted from aliquots of the OMNIgene Gut (OMR−200) tubes for downstream microbiome analysis. DNA yield ranged from 358.5 to 16,660 ng, RNA from 690 to 2010 ng (Fig. 7c). Large variations in yield are attributable to variable stool mass collected between kits.

Stool samples enable various biochemical, immune, and microbiome changes studies. Previous metagenomic assays have found that shannon alpha diversity and richness during long-duration missions to the ISS[62].

### Skin swabs
Body swabs were collected at all timepoints. Samples were collected by swabbing the body region of interest for 30 seconds, then placing the swab in a sterile 2D matrix tube (Thermo Scientific #3710) with Zymo DNA/RNA shield preservative. For the first two swab locations, the oral and nasal cavity, the swab was placed directly on the body after removal from its sterile packaging (dry-swab method; Fig. 8a). For the remaining body locations, the swab was briefly dipped in nuclease-free, DNA/RNA-free water before proceeding (wet-swab method). Eight distinct sites were swabbed with the wet-swab method: post-auricular, axillary vault, volar forearm, occiput, umbilicus, gluteal crease, glabella, and the toe-web space (Fig. 8b). The astronaut microbiome has previously been studied in the forehead, forearm, nasal, armpit, navel, postauricular, and tongue body locations, and changes have been documented during flight. Changes in alpha diversity and beta diversity were documented, as well as shifts in microbial genera[62]. However, the impact of these changes on skin health and immunological health are not well understood.

Acquiring extensive swab samples from the crew skin allows for characterization of the habitat environment, crew skin microbiome adaptations, and interactions with potential human health adaptations resulting from spaceflight exposure. This is very relevant for crew health, considering astronauts become more susceptible to infections during spaceflight missions[63], with the relationship between microbe-host interactions from spaceflight exposure, which may be a causative factor of astronauts immune dysfunction, which is still not well understood.

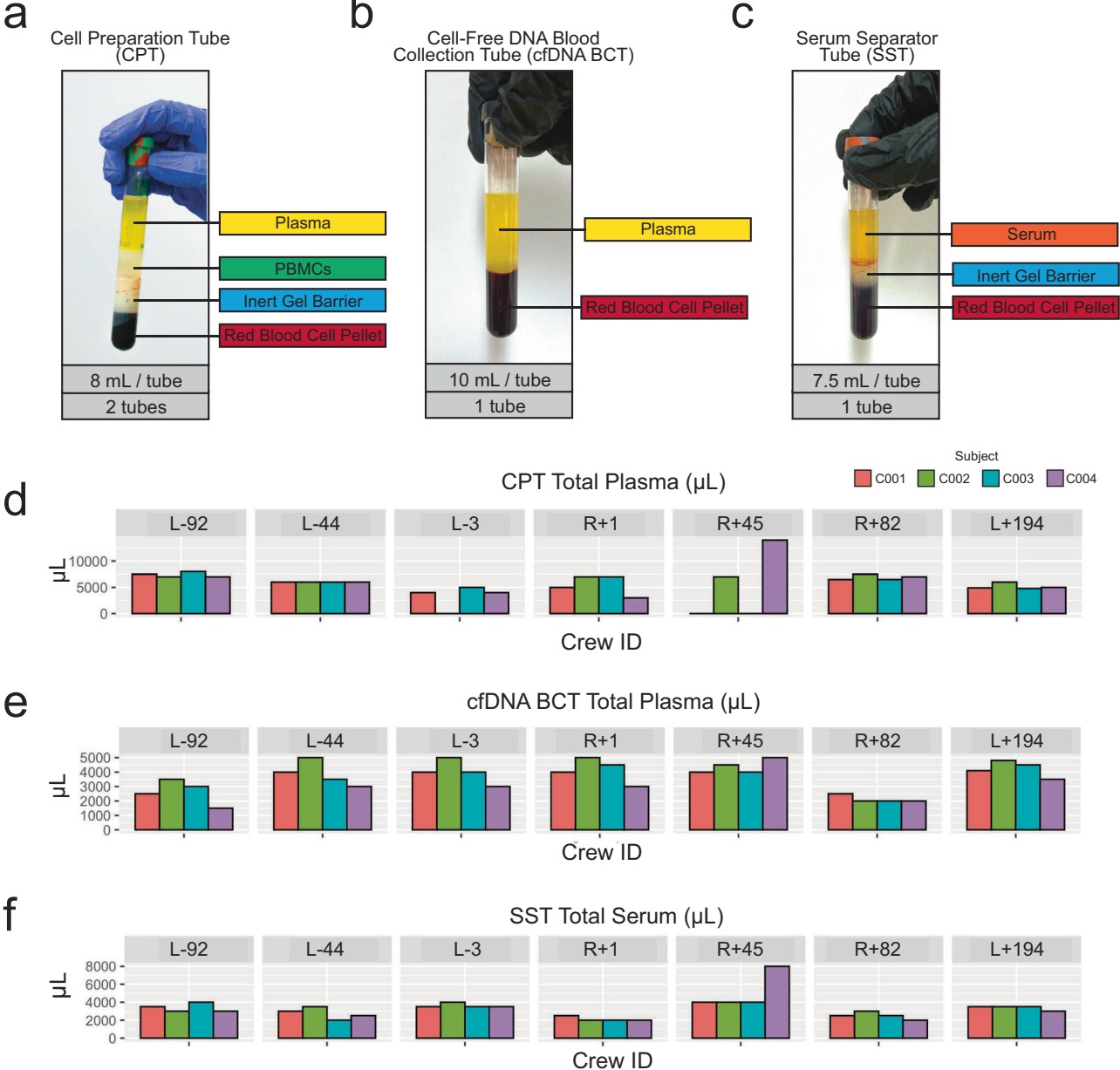

**Fig. 4 | CPT, cfDNA BCT, and SST Yields. a** A spun CPT yields plasma, PBMCs, and a red blood cell pellet. PBMC from each tube were divided into 6 cryovials and viably frozen. Plasma was aliquoted and the pellet was frozen at −20C. **b** A spun cfDNA BCT yields plasma and a red blood cell pellet. Plasma was purified with an additional spin (see Fig. 4a) then aliquoted. The pellet was frozen at −20C. **c** A spun SST yields serum and a red blood cell pellet. Serum was aliquoted and the pellet was frozen at −20C. **d** CPT plasma volumes per timepoint are reported. **e** cfDNA (Streck) BCT plasma volumes per timepoint. **f** SST serum volumes per timepoint. An extra tube was drawn for C004 at R + 45, resulting in a higher serum yield.

## Skin Biopsies

A skin biopsy on the deltoid was obtained from the L-44 and R + 1 timepoint. Biopsies were also collected in advance of a flight to ensure the biopsy site is fully healed before the flight so there is no risk of complication. The wet-swab method was used to collect the skin microbiome before the skin biopsy. The skin biopsies were 3–4 millimeters in diameter and were collected for histology and spatially resolved transcriptomics (SRT) (Fig. 8c). One-third of the sample was stored in formalin and kept at room temperature to perform histology. The remaining two-thirds of the sample was stored in a cryovial and placed at -80 °C for SRT (Fig. 8c). This is the first sample collected from astronauts for spatially resolved transcriptomics. The skin is of high interest due to the inflammation-related cytokine markers such as IL-12p40, IL-10, IL-17A, and IL-18[10,17]

and skin rash's status as the most frequent clinical symptom reported during spaceflight[64].

## Environmental swabs and HEPA filter

Environmental swabs were collected in flight during the F1 and F2 timepoint. Additionally, environmental swabs were collected from the flight simulation capsule at SpaceX headquarters after days of crew training during the L-92 and L-44 timepoints. Environmental swabs were collected using the wet-swab method. Ten environmental swabs were collected per time point at the following locations in the capsule: an ambient air/control swab, the execute button, the viewing dome, the side hatch mobility aid, the lid of the waste locker, the head section of one of the seats, the commode panel, the right and left sides of the control screen, and the g-meter button (Fig. 9a-d). Additionally, the

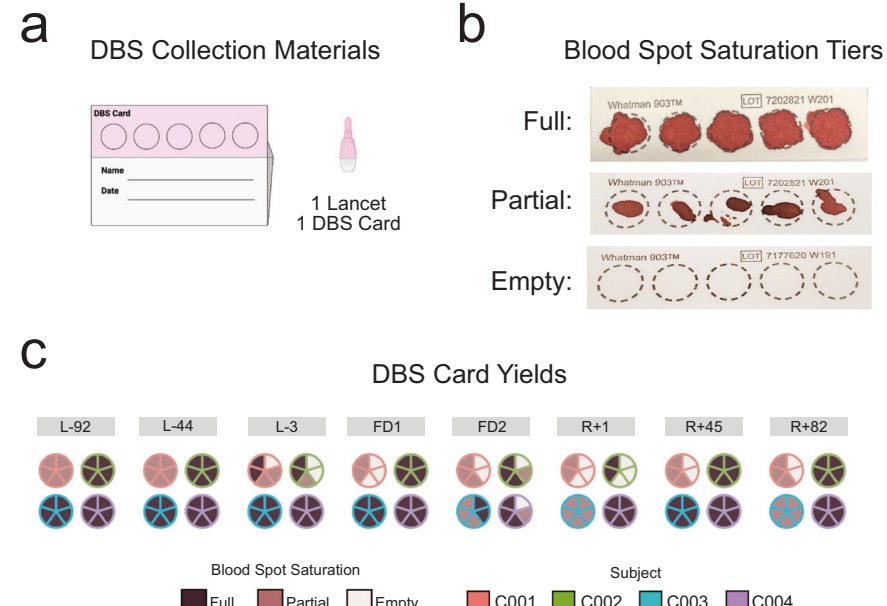

**Fig. 5 | Dried Blood Spot (DBS) Collection Yields. a** DBS cards were collected preflight, during flight, and postflight. There were five spots for blood collection per card. **b** Blood collections varied in saturation level across blood spots and timepoints. These were classified as "full", "partial", and occasionally "empty". **c** DBS card yields per blood spot, per timepoint, and per crew member.

**Fig. 6 | Saliva Collections. a** DNA, RNA, and protein yields from the OMNIgene Oral kits. **b** Volume of crude saliva collected per timepoint.

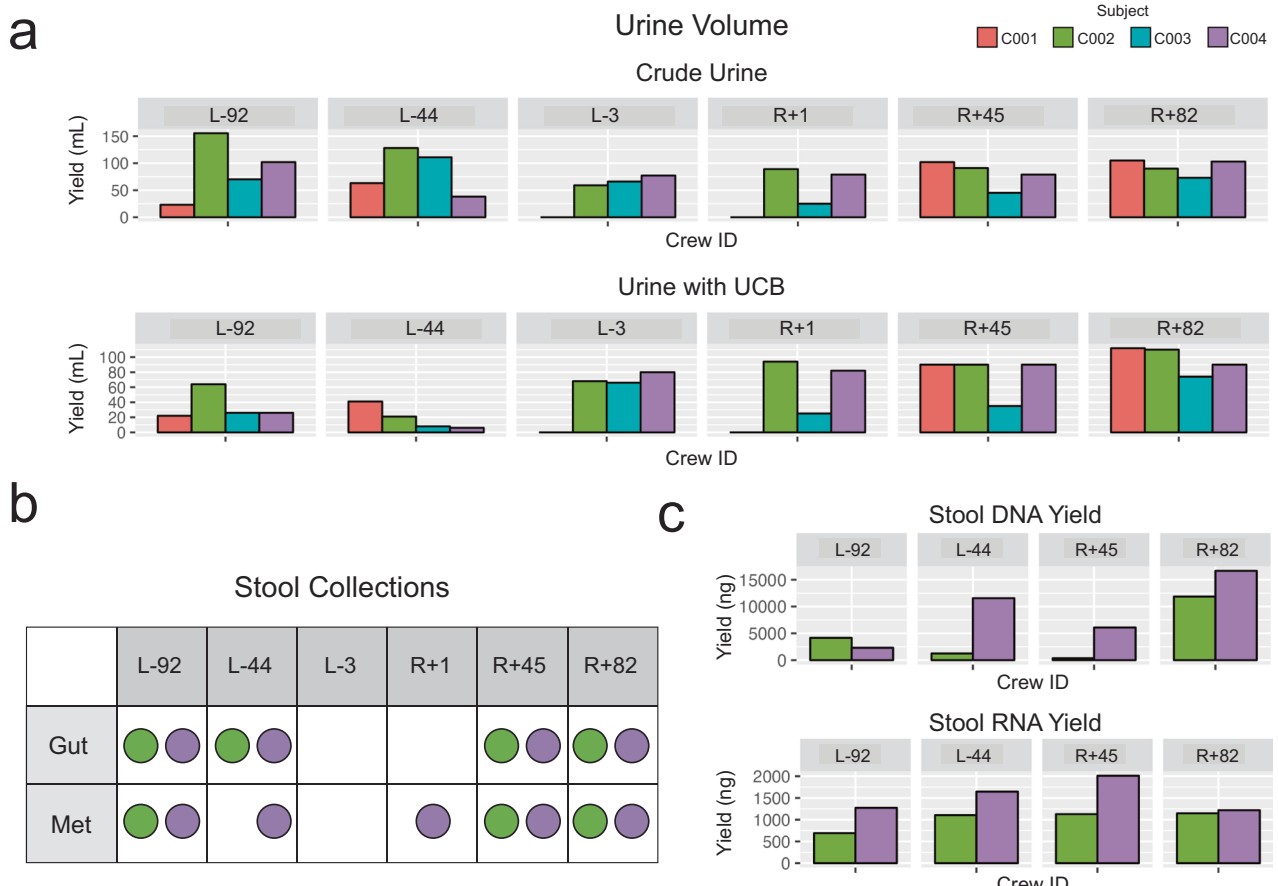

**Fig. 7 | Urine and Stool Sample Collections. a** Urine volumes per timepoint. Volumes are reported for both crude urine and urine preserved with Zymo urine conditioning buffer (UCB). **b** Timepoints that stool tubes were collected. "Gut" tubes are OMNIgene•GUT tubes for microbiome preservation. "Met" tubes are OMNImet•GUT tubes for metabolome preservation. **c** Stool "Gut" tube DNA and RNA extraction quantities.

spacecraft's high-efficiency particulate absorbing (HEPA) filter was acquired post-flight. This filter was cut into 127 rectangular pieces (1.2" x 1.6" x 4") and stored at −20 °C.

Previous microbial profiling of spacecraft environments has revealed that equipment sterilized on the ground becomes coated in microbial life in space due to interactions with crew and the introduction of equipment that has not undergone sterilization[65]. Subsequent microbial monitoring assays performed on the ISS have detected novel, spaceflight-specific species on the ISS[66]. Once in space, surface microbes are subject to the unique microgravity and radiation environment of flight, which will influence evolutionary trajectory. The potential impact of this influence on pathogenesis is a concern for long-duration space missions, especially given that changes in host-pathogen interactions may also be affected during spaceflight[67].

## Discussion

We report here on biospecimen samples collected from the SpaceX I4 Mission, the most comprehensive human biological specimen collection effort performed on an astronaut cohort to date. The extensive archive of biospecimens included venous blood, dried blood spot cards, saliva, urine, stool, microbiome body swabs, skin biopsies, and environmental capsule swabs. The study objective was to establish a foundational set of methods for biospecimen collection and banking on commercial spaceflight missions suitable for multi-omic and molecular analysis. Biospecimens were collected to enable comprehensive, multi-omic profiles, which can then be used to develop molecular catalogs with higher resolution of human responses to spaceflight. Select, targeted measures in clinical labs (CLIA) were also

performed immediately after sample collection (CBC, CMP), and samples and viable cells were preserved in a long-term Cornell Aerospace Medicine Biobank, such that additional assays and measures can be conducted in the future.

Great care was put into the standardization of sample processing between field laboratories in Hawthorne, Cape Canaveral, and Houston. Despite our best efforts to control for all preanalytical variables[68], there are some inconsistencies in sample processing that were unavoidable during this mission. Of note, there was no access to cold stowage during flight, causing all samples to be stowed at room temperature until they arrived back on Earth. Of the three sample types collected during flight, two were stable at room temperature (skin/capsule swabs stored in DNA/RNA shield and capillary blood collected on dried blood spot cards). The third (saliva samples), were not stored in a preservative and, for this reason, will have a distinct profile compared to other timepoints. This limits the context in which these samples will be useful, but will still provide biological insights in more targeted biological studies. We can imagine situations in the future where there will be tradeoffs between standardization of sample collection with previous missions and mission-specific constraints influencing the selection of biospecimen collection methods.

However, there are several reasons why rigorous biospecimen collection methods for commercial and private spaceflight missions must be developed, which are scalable and translational across populations, missions, and mission parameters. First, little is known about the biological and clinical responses that occur in civilians during and after space travel. While professional astronauts are generally young, healthy, and extensively trained, civilian astronauts have

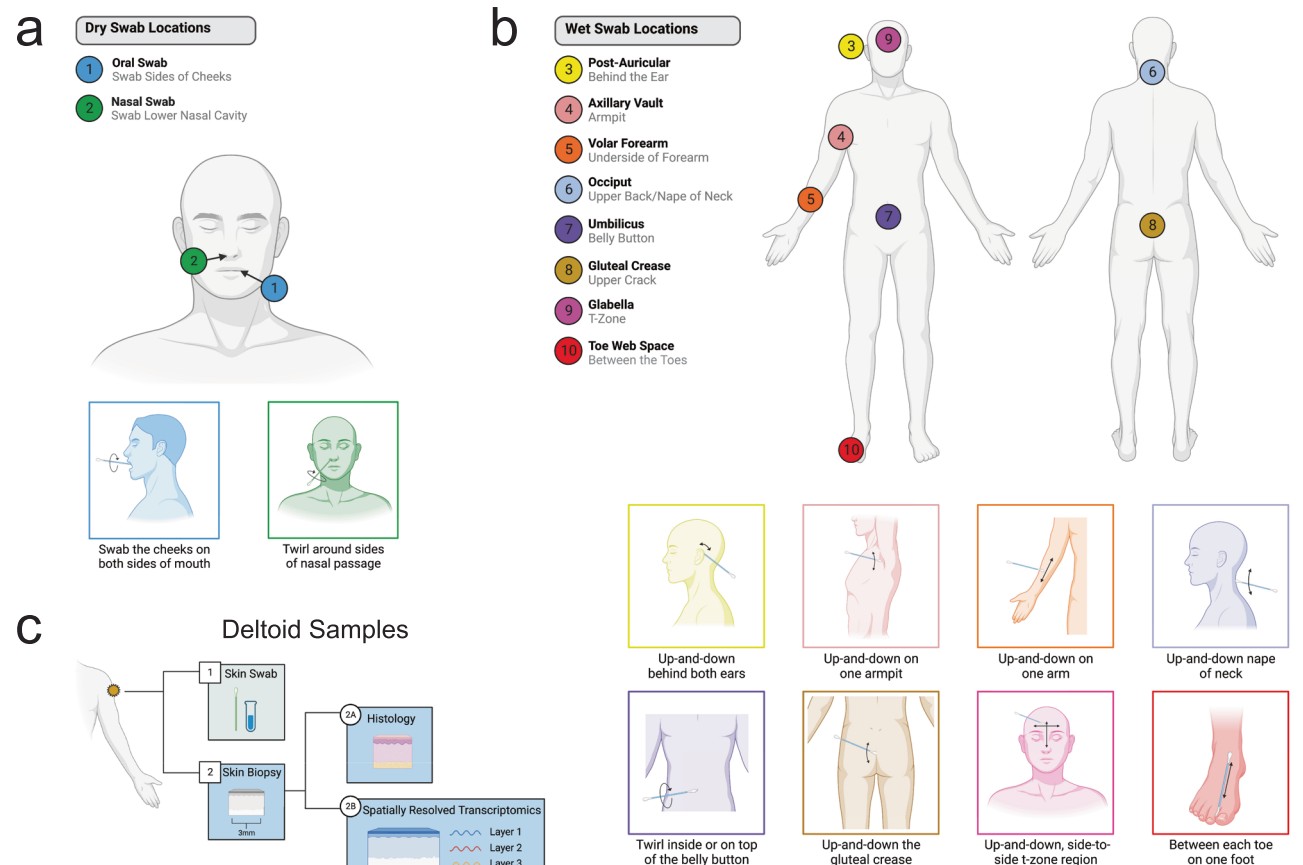

**Fig. 8 | Skin Collection Locations and Sample Types. a** Dry swabs were collected from two body locations. **b** Wet swabs were collected from eight body locations. **c** Swabs were collected from the deltoid region. Immediately after, 3- or 4-mm skin biopsies were collected from the same area and divided for histology and spatially resolved transcriptomics.

been, and likely will be, far more heterogeneous. They will possess a variety of phenotypes, including older ages, different health backgrounds, and greater medication use, and may experience different medical conditions, risks, and comorbidities. Careful molecular characterization will be beneficial for the development of appropriate baseline metrics and countermeasures and, therefore, beneficial for the individual spaceflight experience. In the future, such analyses may enable precision medicine applications aimed at optimizing countermeasures for each individual astronaut who enters and returns safely from space[69,70].

Second, multi-omic studies inherently present a large number of measurements within a small set of subjects. These high-dimensional datasets present numerous potential challenges with regard to the amplification of noise, risk of overfitting, and false discoveries[71]. At all times, scientists engaged in multi-omic analyses must take special care that true biological variance is what has been measured. The introduction of experimental variance through the progression from sample collection, transport, and storage, to sequencing and analysis can introduce artifacts of variance that render the detection of true biological variance and interpretation of results more difficult. For this reason, tight adherence to experimental controls or annotation at every step of the experimental condition is crucial. Careful annotation allows for the assignment of class variables in post hoc analysis. Among such applications are the attempt to detect batch effects or determine the impact of variations in temperature (collection, storage, or transport)[72].

The necessary means to address experimental variance are longitudinal sampling and specimen aliquoting. Longitudinal sampling (i.e. collecting numerous serial samples from each test condition) from

pre-flight, in-flight, and post-flight allows for greater statistical power when assessing changes attributable to spaceflight. In addition, each sample collected should be divided upon collection into multiple aliquots. This better assures that freeze-thaw cycles can be avoided in the analysis stage, as freeze-thaw events can introduce considerable experimental variance depending on the molecular class being measured. Maintaining all samples at their optimal storage temperature at all times, typically -80 °C or lower (e.g. liquid nitrogen for cells), is crucial[73]. Special attention must be given to how the collection and storage methods in-flight vary in relation to the conditions on Earth. Spaceflight presents considerable differences in the operating environment, where ground conditions are far easier to control than flight. In practice, this may limit the types of samples that can be collected during flight.

Third, rigorous methods must be developed and followed to pursue comparisons across missions with varying design parameters. In this consideration, there is an argument for the development of specimen collection, transport, storage, processing, analysis, and reporting standards. At the same time, this must be balanced with the flexibility required for innovation since standards can sometimes limit advancement in methodology. In the present study, common methods were used for the I4 and the Polaris Dawn and Axiom-2 missions. However, selected methods may require optimization for Polaris Dawn-like missions, to increase the yields during sample processing and adapt to unique parameters imposed by the anticipated spacewalk (extravehicular activity; EVA). Moreover, within standards or best practices, unique research for each mission may require alteration of previously successful methods. With these considerations in mind, we must balance methodology

a
Environmental Swab Locations

| ID | Location | Description |
|---|---|---|
| 0 | Control Swab | Dampened swab; hold in air for 30 seconds. |
| 1 | Execute Button | Physical button on the control panel. |
| 2 | Viewing Dome | Bottom rim of the cupola, towards the crew entrance. |
| 3 | Side Hatch Mobility Aid | Side of the spacecraft. View from the camera angle in Figure 6b. |
| 4 | Lid of Waste Locker | Quarter turn screws and surface of panel was swabbed. Located towards the bottom-right floor of Figure 6b. |
| 5 | Seat 2 | Upper section by the head area was swabbed. |
| 6 | Commode panel | Quarter turn screws and bottom part of panel was swabbed. Located at the top of Figure 6b. |
| 7 | Control Touch Screen | Left side of the screen. |
| 8 | Control Touch Screen | Right side of the screen. |
| 9 | G-meter Button | Under an acrylic barrier. |

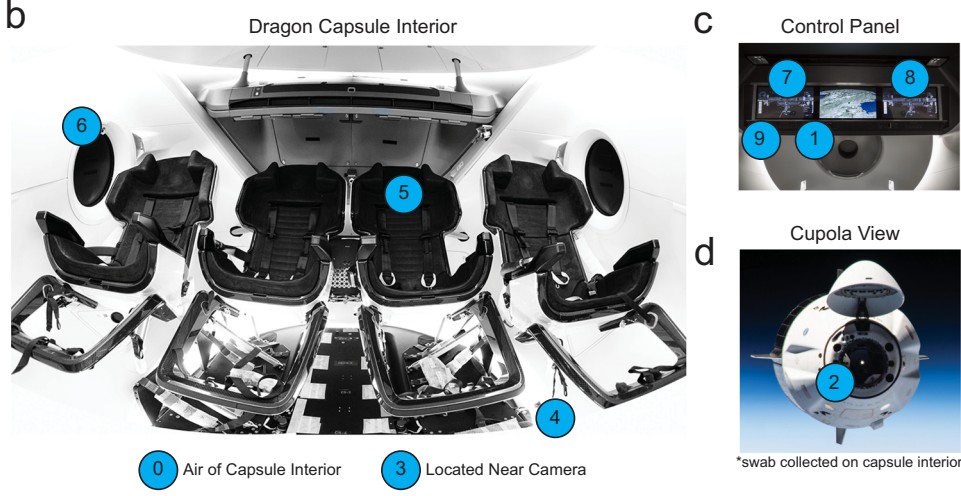

**Fig. 9 | Capsule Swab Locations. a** Swab locations, descriptions, and label IDs. **b** Interior view of the SpaceX Dragon capsule. **c** View of the control panel located above the middle seats in the Dragon capsule. **d** View of the cupola (viewing dome) region from the outside. The rim of the dome was swabbed from the inside (ID 10).

standardization with advances in methodology options and mission-specific objectives.

As the commercial spaceflight sector gains momentum and more astronauts with different health profiles and backgrounds have access to space, comprehensive data on the biological impact of short-duration spaceflight is of paramount importance. Such data will further expand our understanding and knowledge of how spaceflight affects human physiology, microbial adaptations, and environmental biology. The use of integrative omics technologies for civilian astronauts has revealed novel responses across genomics, proteomics, metabolomics, microbiome, and transcriptomics measures[74–83]. Creating multi-omic datasets from spaceflight studies on astronaut cohorts will further advance our understanding, inform future mission planning, and help discover what appropriate countermeasures can be developed to minimize future risk and enhance performance.

Validating sample collection methodologies initially in short-duration commercial spaceflight is a key step for future human health research in long-duration and exploration-class missions to the Moon and beyond. To help meet these challenges, we have established the SOMA protocols, which detail standard multi-omic measures of astronaut health and protocols for sample collection from astronaut cohorts. Although the all-civilian I4 crew pioneered the first use of the SOMA protocols, the methodology outlined here is robust and generalizable, making it applicable to future astronaut crews from any commercial mission provider (e.g., SpaceX, Axiom

Space, Sierra Space, Blue Origin) or space agencies (NASA, ESA, JAXA, ROSCOSMOS). Furthermore, the SOMA banking, sequencing, and processing methods are a springboard for continuing biospecimen analysis and expanding our knowledge of multi-omic dynamics before, during, and after human spaceflight missions, providing a molecular roadmap for crew health, medical biometrics, and possible countermeasures.

## Methods
### Venous blood draw
Venipuncture was performed on each subject using a BD Vacutainer® Safety-Lok™ blood collection set (BD Biosciences, #367281) and a Vacutainer one-use holder (BD Biosciences, 364815). The puncture site was located near the cubital fossa and was sterilized with a BZK antiseptic towelette (Dynarex, Reorder No. 1303). Blood was collected into 1 serum separator tube (SST, BD Biosciences: #367987, Lot: #1158449, #1034773), 2 cell processing tubes (CPT, BD Biosciences: #362753, Lot: #1133477, #1012161), 1 blood RNA tube (bRNA, PAXgene: #762165, Lot: #1021333), 1 cell-free DNA BCT (cfDNA BCT, Streck: #230470, Lot: #11530331), and 4 K2 EDTA blood collection tubes (BD Biosciences, #367844, Lot: #0345756) per crew member per time point. For samples collected in Hawthorne, blood was drawn at SpaceX headquarters, then immediately transported to USC for processing. Samples collected at Cape Canaveral were processed on-site.

## Blood tube processing

For processing, serum separator tubes (SST) were centrifuged at 1300x *g* for 10 minutes. 500uL aliquots of serum were aliquoted into 1 mL Matrix 2D Screw Tubes (ThermoFisher, 3741-WP1D-BR) and stored at -80 °C. SST tubes were recapped and stored at −20 °C to preserve the red blood cell pellet.

Cell processing tubes were centrifuged at 1800xg for 30 minutes. Plasma was aliquoted into 1 mL Matrix 2D Screw Tubes and stored at -80 °C. 5 mL of 2% FBS (ThermoFisher, #26140079) in PBS (Thermo-Fisher, #10010023) was added to the CPT tube to resuspend PBMCs. PBMC suspension was transferred to a clean 15 mL conical tube. The total volume was brought to 15 mL with 2% FBS in PBS. The tube was centrifuged for 15 minutes at 300x *g*. Supernatant was discarded. PBMCs were resuspended 6 mL of 10% DMSO (Millipore Sigma, #D4540-500mL) in FBS. 1 mL of PBMCs were moved to 6 cryogenic vials (Corning, #8672). Cryovials were placed in a Mr. Frosty™ (ThermoFisher, #5100-0001) and stored at -80 °C. CPTs were recapped and stored at −20 °C to preserve the red blood cell pellet. There was one outlier to this sample collection, the R + 194 timepoint, where CPT sodium citrate tubes were used (BD Biosciences, Cat no. 362760).

cfDNA BCTs were centrifuged at 300xg for 20 minutes. Plasma was transferred to a 15 mL conical tube. Plasma was centrifuged 5000xg for 10 minutes. 500uL aliquots of plasma were aliquoted into 1 mL Matrix 2D Screw Tubes and stored at -80 °C. cfDNA BCTs were recapped and stored at −20 °C to preserve the red blood cell pellet.

PAXgene blood RNA tubes were processed according to the manufacturer's instructions. Briefly, tubes were left upright for a minimum of 2 hours before freezing at −20 °C. For RNA extraction, tubes were thawed and processed with the PAXgene blood RNA kit (Qiagen, #762164).

## Extracellular Vesicles and Particles (EVPs) Isolation

One 4 mL K2 EDTA tube was shipped on ice overnight to WCM for processing. Blood was centrifuged at 500x *g* for 10 minutes, then plasma was transferred to a new tube and centrifuged at 3000 x g for 20 minutes, and the supernatant was collected and stored at -80 °C for EVP isolation. Plasma volumes ranged between 0.6 - 1.7 ml. Plasma was later thawed for downstream processing, when concentrations were measured. Plasma samples were thawed on ice, and EVPs were isolated by sequential ultracentrifugation (Hoshino et al., 2020). Samples were centrifuged at 12,000x *g* for 20 minutes to remove microvesicles, then EVPs were collected by ultracentrifugation in a Beckman Coulter Optima XE or XPE ultracentrifuge at 100,000x *g* for 70 minutes. EVPs were then washed in PBS and pelleted again by ultracentrifugation at 100,000x *g* for 70 minutes. The final EVP pellet was resuspended in PBS.

## Dried blood spot (DBS)

Crew members warmed their hands and massaged their finger towards the fingertip to enrich blood flow toward the puncture site. The puncture site was sterilized using a BZK antiseptic towelette (Dynarex, Reorder No. 1303). Skin was punctured using a contact-activated lancet (BD Biosciences, #366593) or a 21-gauge needle (BD Biosciences, #305167), depending on crew member preference. Capillary blood was collected onto the Whatman 903 Protein Saver DBS cards (Cytiva, #10534612). Blood was transferred by touching only the blood droplet to the surface of the DBS card. DBS cards were stored at room temperature with a desiccant pack (Cytiva, #10548239).

## Saliva

To collect crude saliva, crew members uncapped and spit into a sterile, PCR-clean, 5 mL screw-cap tube (Eppendorf, 30122330). Crew spit repeatedly until at least 1 mL was collected. Saliva was transported to a sterile flow hood and separated into 500uL aliquots. Aliquots were frozen at -80 °C. To collect preserved saliva, crew members used the OMNIgene ORAL kit (OME-505). Crew members spit into the kit's tube

until they reached the fill line. The tube was re-capped, which released the preservative liquid. Tubes were inverted to mix the saliva and preservative before being placed at −20 °C for storage. After all time-points were collected, DNA, RNA, and protein were extracted using the AllPrep DNA/RNA/Protein kit (Qiagen, #47054). Sample concentrations were measured with Qubit high sensitivity dsDNA and RNA platform. Proteins were quantified with the Pierce™ Rapid Gold BCA Protein Assay Kit (Thermo Scientific, #A53225) on Promega GloMax Plate Reader.

## Urine

Crew members urinated into sterile specimen containers (Thermo Scientific, #13-711-56). The container was stored at 4 C until it was prepared for long-term storage. To prepare urine samples for long-term storage, urine was aliquoted into 1 mL, 15 mL, and 50 mL tubes. Half of the urine was immediately placed at −80 °C. The other half had urine conditioning buffer (Zymo, #D3061-1-140) added to the sample before placing in the -80 °C freezer.

## Stool collection

Crew members isolated a stool sample using a paper toilet accessory (DNA Genotek, OM-AC1). Stool was transferred into and OMNIgen-e•GUT tube (DNAgenotek, OMR−200) and an OMNImet•GUT tube (DNA Genotek, ME−200). Tubes were placed at -80 °C for long-term storage. For nucleic acid extraction, 200uL of each tube was allocated for DNA extraction with the QIAGEN PowerFecal Pro kit and 200uL was allocated to RNA extraction with the QIAGEN PowerViral kit. The remaining sample was split into 500uL aliquots and re-stored at -80 °C.

## Swab collection

Crew members put on gloves and remove a sterile DNA/RNA swab (Isohelix, SK-4S) from its packaging. For collection of the post-auricular, axillary vault, volar forearm, occiput, umbilicus, gluteal crease, glabella, toe web space, and capsule environment regions, swabs were dipped in nuclease-free water (this step was skipped for oral and nasal swabs) for ground collections. For in-flight collections, HFactor hydrogen infused water was used in place of nuclease-free water. Each body location was swabbed for 30 seconds, using both sides of the swab. Swabs were then placed in 1 mL Matrix 2D Screw Tubes containing 400uL of DNA/RNA Shield (Zymo). The tip of the swab was broken off so that only the swab tip was stored in the Matrix 2D Screw Tube. Tubes were stored at 4 C.

## Skin biopsies

Skin biopsies were performed on the deltoid region of the arm. Each site was prepared by application of ChloraPrep and anesthesia was induced with administration of 1% lidocaine with 1:100,000 epinephrine. A trephine punch was used to remove a 3- or 4-mm diameter piece of skin. The resected piece was cut into approximately $1/3$ and $2/3$ sections. The smaller piece was added to a formalin-filled specimen jar. The larger piece was placed in a cryovial and stored at -80 °C. Surgical defects were closed with 1 or 2 5-0 or 4-0 nylon sutures.

## HEPA filter

HEPA Filter was taken apart and sectioned under a chemical hood to avoid contamination. The filter contained two parts, an activated carbon component and a HEPA filter. The activated carbon component was discarded and the filter was sectioned using a sterile blade. Sections were placed in individual specimen containers and stored at −20 °C.

## Human subjects research

All subjects were consented and samples were collected and processed under the approval of the IRB at Weill Cornell Medicine, under Protocol 21-05023569.

**Manuscript preparation**
Figures 1c, 2c, 3, 5a, 8 were created with BioRender.com released under a 'Creative Commons Attribution-NonCommercial-NoDerivs 4.0 International license'. Plots were generated in R using ggplot2.

**Reporting summary**
Further information on research design is available in the Nature Portfolio Reporting Summary linked to this article.

## Data availability
Images of the Inspiration4 and Dragon Capsule are available from the SpaceX Flickr Account (https://www.flickr.com/people/spacex/). This was the source image data used to label the Dragon swab location in Fig. 9.

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

## Acknowledgements
We thank the Scientific Computing Unit (SCU) at WCM and the Geno-mics, Epigenomics, and Biorepository Cores. CEM thanks the NIH

 

(RO1MH117406, RO1ES032638) and NASA (NNX14AH50G, NNX17AB26G, 80NSSC22K0254, NNH18ZTT001N-FG2, 80NSSC23K0832), the LLS (MCL7001-18, LLS 9238-16, 7029-23), as well as Igor Tulchinsky and the WorldQuant Foundation, the GI Research Foundation (GIRF), the Radvinsky/Chudnovsky family. EGO thanks NASA BPS (80NSSC23K0832). We thank JJ Hastings for protocol work. JK thanks MOGAM Science Foundation and was supported by the Basic Science Research Program through the National Research Foundation of Korea (NRF) funded by the Ministry of Education (RS–2023-00241586). JK acknowledges Boryung for their financial support and research enhancement ground, provided through their Global Space Healthcare Initiative, Humans In Space, including mentorship and access to relevant expert networks. We also thank Jennifer Conrad for CPT photography.

## Author contributions

C.E.M., J.M., M.M. led study conceptualization. C.E.M., J.M., M.M., E.G.O., S.M. developed study methodology. C.E.M., M.Y., J.M., and D.L. acquired resources and funding. C.E.M., J.M., and E.G.O. administered the project. E.G.O., K.R., B.T.T., R.K., V.O., S.M., and R.D.G. performed off-site sample collection protocols. J.K., I.M., L.P., N.D., D.N., J.W.H., J.S.G.M., J.P., M.M.S., and A.S.K. performed sample processing at Weill Cornell Medicine. C.E.M. and E.E.A. led institutional review board (IRB) protocol development. E.G.O., M.M., and L.I. developed figures. E.G.O. and J.C.S. drafted the manuscript. S.A.N., C.M.S., M.A.S., and B.S. reviewed and edited the manuscript.

## Competing interests

CEM is co-Founder of Cosmica Biosciences. BTT is compensated for consulting with Seed Health and Enzymetrics Biosciences on microbiome study design and holds an ownership stake in the former. CMS, JCS, and MAS hold shares in Sovaris Holdings, LLC. MY is the founder and president of CanTraCer Biosciences Inc. Authors not listed here do not have competing interests.

## Additional information

[1]Department of Physiology and Biophysics, Weill Cornell Medicine, Cornell University, New York, NY, USA. [2]The HRH Prince Alwaleed Bin Talal Bin Abdulaziz Alsaud Institute for Computational Biomedicine, Weill Cornell Medicine, New York, NY, USA. [3]BioAstra, Inc, New York, NY, USA. [4]Center for STEM, University of Austin, Austin, TX 78701, USA. [5]Department of Stem Cell Biology and Regenerative Medicine, Keck School of Medicine, University of Southern California, Los Angeles, CA, USA. [6]Department of Pharmacology, University of Maryland School of Medicine, Baltimore, MD 21201, USA. [7]Space Exploration Technologies Corporation, Hawthorne, CA, USA. [8]Sovaris Aerospace, Boulder, Colorado, USA. [9]Advanced Pattern Analysis & Human Performance Group, Boulder, Colorado, USA. [10]Children's Cancer and Blood Foundation Laboratories, Departments of Pediatrics and Cell and Developmental Biology, Drukier Institute for Children's Health, Weill Cornell Medicine, New York, NY, USA. [11]Meyer Cancer Center, Weill Cornell Medicine, New York, NY 10065, USA. [12]Department of Molecular Biology and Biotechnology, Center of Systems Biology, Biodiversity and Bioresources, Faculty of Biology and Geology, Babes-Bolyai University, Cluj-Napoca, Romania. [13]Florida State University, College of Education, Health, and Human Sciences, Department of Health, Nutrition, and Food Sciences, Tallahassee, FL, USA. [14]Department of Systems Engineering, Colorado State University, Fort Collins, Colorado, USA. [15]Hematology and Oncology Division, Weill Cornell Medicine, New York, NY, USA. [16]Department of Dermatology, Weill Cornell Medicine, New York, NY, USA. [17]Department of Neuroscience, King Faisal Specialist Hospital & Research Centre, Jeddah, Saudi Arabia. [18]The Feil Family Brain and Mind Research Institute, Weill Cornell Medicine, New York, NY 10021, USA. [19]WorldQuant Initiative for Quantitative Prediction, Weill Cornell Medicine, New York, NY 10021, USA. ✉e-mail: chm2042@med.cornell.edu

