## [Peer Review File · Nature Communications]

Collection of Biospecimens from the Inspiration4 Mission Establishes the Standards for the Space Omics and Medical Atlas (SOMA)REVIEWER COMMENTS

Reviewer #1 (Remarks to the Author):

Review

The manuscript describes in general terms the biobanking protocol for the Inspiration4 Mission. Overall, nice overview and graphical displays.

Please, consider the comments below to improve the paper. I do not want to be pedantic as the authors already have done a great job in describing details. But a bit more clarification is needed in order for future readers of the future papers based on these samples to evaluate and interpret the quality of the samples.

Abstract:

Write out the full word for L, FD, and R. I guess you mean launch (L), FD(?), return (R)

Replace "venous blood" with the actual primary collection: ex. Venous whole blood, plasma, and serum.

Provide the time frame with collection year(s) and add total number of people and total number of samples.

Manuscript:

You have presented a nice geographical map of the collection process. But this also raises concerns about the transport of the material, which is not clear from the manuscript. Transport of biospecimens should be described in detail (when, when in the process/timeline, which temperature, which material for transport (dry ice, other?), which courier, flight/train/car to transport long-distance, time in transport etc.).

A critical and realistic reflection on how biological and pre-analytical variables (including transport conditions) could have impacted specimen stability and quality. For reference: PMID: 25979952.

Did you notice any degree of visual hemolysis in the samples or measure hemolysis?

The authors mention that standardization is critical for the later utilization and interpretation of the samples. I agree. Which standards were followed (CLIA, ISBER, other)?

I like the overview in Table 1. But Table 1 is massive and could preferably be divided into smaller focused samples based on the sample type. You can also try to reduce the white space and adjust the fonts and line spacing as simple measures. The duration is not only listed in days, but also years and months; however, for comparison across studies, it would help to have at least the duration in days for all studies. Somewhere in the table you have whole blood and serum in column one, and hematology in column 2, but hematology can only be measured on whole blood. In some of the rows for column 3 you have mentioned locations instead of participant N. It is not clear if each line represent one space mission, or the same space mission can cover several lines; therefore it would be great to add the name of mission and the year of the mission. Sometimes you mention sex and sometimes not (please be consistent or write if data on this is missing). Sometimes in column 3, you write "astronauts", but aren't all these studies for astronauts?

For all tables provide a legend describing ALL the abbreviations.

It is not clear from table 2, column 3, which measurements are already measured on fresh samples, which were measured on frozen samples, and which are intended to be measured in the future on biobanked samples. You write "biobanking" for some but not all.

Table 3 should also include genomic DNA and RNA I guess.

Describe the geographical locations of the processing and the biobanking. And was it centralized or decentralized?

Describe time from sampling to processing and freezing for all samples. Were some sample types frozen before processing? What was the temperature of the samples pre-processing (ambient temperature, refrigerator, or frozen) after collection? When did the aliquoting take place (duration from the primary collection and what was the temperature of samples between collection and aliquoting)?

Some samples have a limited time window from primary tube collection to processing to ensure the stability of analytes. Please add the time windows for the samples.

Add the specific color of the "stopper" (the exact color is provided by the company) and colors also indicate type of material.

Write out BD for the vacutainer type.

Which grade of DBS did you use?

Who among the authors was responsible for the standard operating procedures and for ensuring that the collection, processing, aliquoting, and storage were performed according to the SOPs (quality assurance)?

Was any of the authors or associated personnel trained in biobanking procedures (there are various certificates and courses in this, some endorsed by ISBER).

Have you planned how to prospectively control/check the quality of the samples? Some studies set aside certain aliquots for this, and at regular intervals check for some biomarkers, so a degradation profile can be estimated.

It is not quite clear if any biomarkers were measured on fresh samples. If so, which?

Be aware that some of the biomarkers you mention (such as cortisol and cytokines) have extremely strict requirements for collection (time of day for cortisol, time on the bench for cytokines) and are sensitive to long-term storage (cytokines).

I am not sure exactly where the biobank is. Can you clarify?

All abbreviations should be mentioned at least once in full writing.

Were the lab(s) CLIA certified?

You use the term "established the standard" in the title. What exactly do you mean by that?

Standardization can mean many things in the clinical laboratory: harmonization (that procedures are the same), but it can also refer to reference measurement procedures and reference materials. Usually, it is governmental agencies or larger organizations that set the standards for the clinical laboratory. Thus, please clarify what you mean by standardization in the title and in the manuscript.

What time of the day were samples taken? And fasting/non-fasting considerations?

Reviewer #2 (Remarks to the Author):

The study by Overbey and co-workers has a specific focus on collection and banking of biospecimens within an exceptional context, which is the SpaceX Inspiration4 mission, with the aim of studying the impact of spaceflight on the human body. The novelty of the study certainly resides in the exceptional source of these samples, however the impact these collection may have on further studies is less clear.

The collection is well described in itself, even with a pleonastic amount of details.

On the other side the study is mainly based on a description of standard parameters related to quality of samples and their analytes (testing of DNA, RNA, proteins, metabolites, and other biomolecules).

Not all of the samples were collected during the flight. Histological samples in the form of skin biopsies were collected at ground timepoints. It is unclear how skin biopsies can be informative and exploited for instance for spatial genomic analyses (such as spatially resolved, single-molecule, and single-cell assays).

Although the effort in the meticulous collection should be praised, the study is very descriptive in nature and the data reported do not provide a significant impact on our knowledge.

It is also questionable whether there will be a systematic collection of histological samples from personnel for these flights that will be used for research studies. We cannot exclude this will be the case, and in this respect this precise indication on best practice of collection of samples will be useful, yet not ground-breaking.

We can foresee that the most relevant biospecimens that will be systematically collected from these units of personnel will be blood and urine to monitor health status of individuals involved in these missions. Again, the paper offers suggestions for good practice in this respect without leading to major advances.

We thank the reviewers for their comments on our manuscript and for the opportunity to provide revisions. In response to the points raised, we have we focused on the following areas for improvement:

- (1) **Clarifications to detail the work we performed.** We clarified which steps were performed in various field laboratories vs. back in our central laboratory location at Weill Cornell Medicine (WCM). We added Extended Data Figure 1 to highlight when assays were performed on fresh samples at a field laboratory location. We added sections and updated the tables to provide full context of the information in the manuscript. We also added a table with the total number of sample aliquots collected across the duration of the study in field laboratories that were shipped back to WCM. Furthermore, we added additional information about shipping logistics and, in our response below, clarified where we are compliant with key sample collection and biobanking recommendations from the International Society for Biological and Environmental Repositories (ISBER).
- (2) **Broader context for the methods and data.** We have elaborated below on the broader impact of the work presented in this paper. Without a connection to the other papers submitted, understandably the paper leaves readers curious about analyses and results. To provide additional context for reviewers, in addition to our comments below, we are sending the Park *et al.* paper, “Spatial multi-omics of human skin reveals KRAS and inflammatory responses to spaceflight”, under review at Nature Communications and the Overbey *et al.* paper, “The Space Omics and Medical Atlas (SOMA): A comprehensive data resource and biobank for astronauts”, under review at *Nature* to assist reviewers in their assessment of the impact and scope of our work.

In addition to these three points, we responded to each of the Reviewer’s comments one-by-one in blue below, citing specific passages from the text when necessary.

Thank you for your consideration, and we look forward to your response.

Dr. Christopher Mason
Weill Cornell Medicine

REVIEWER COMMENTS

Reviewer #1 (Remarks to the Author):

Review

The manuscript describes in general terms the biobanking protocol for the Inspiration4 Mission. Overall, nice overview and graphical displays.

Please, consider the comments below to improve the paper. I do not want to be pedantic as the authors already have done a great job in describing details. But a bit more clarification is needed in order for future readers of the future papers based on these samples to evaluate and interpret the quality of the samples.

Abstract:

Write out the full word for L, FD, and R. I guess you mean launch (L), FD(?), return (R)
Replace "venous blood" with the actual primary collection: ex. Venous whole blood, plasma, and serum.

Provide the time frame with collection year(s) and add total number of people and total number of samples.

We have made the clarification revisions and write out the full words for "L" (launch), "FD" (Flight Day), and "R" (return) to improve clarity and ensure better comprehension for readers. We acknowledge your suggestion to replace the term "venous blood" with more precise descriptions such as "venous whole blood," "plasma," and "serum" wherever applicable. Additionally, we now include the timespan and collection year(s) to provide a clear reference for our study period, as well as the number of participants and samples in our revised manuscript. We agree that providing these specific terms will enhance the accuracy and completeness of our abstract.

Manuscript:

You have presented a nice geographical map of the collection process. But this also raises concerns about the transport of the material, which is not clear from the manuscript. Transport of biospecimens should be described in detail (when, when in the process/timeline, which temperature, which material for transport (dry ice, other?), which courier, flight/train/car to transport long-distance, time in transport etc.).

Details about sample transfer to Weill Cornell Medicine have been added to the manuscript in our discussion of Fig 1C. All samples were processed within 16h, and details are now included that give more specific time frames.

A critical and realistic reflection on how biological and pre-analytical variables (including transport conditions) could have impacted specimen stability and quality. For reference: PMID: 25979952.

Careful planning was performed to standardize sample processing between field laboratories to limit the impact of preanalytical variables. However, there is one outlier during the in-flight sample collection, which we now highlight in the discussion of the paper:

"Great care was put into the standardization of sample processing between a handful of field laboratories in Hawthorne, Cape Canaveral, and Houston. Despite our best efforts to control for all preanalytical variables(Ellervik and Vaught 2015), there are some inconsistencies in sample processing that were unavoidable during this mission. Of note, there was no access to cold stowage during flight, causing all samples to be stowed at room temperature until they arrived back on Earth. Of the three sample types collected during flight, two were stable at room temperature (skin/capsule swabs stored in DNA/RNA shield and capillary blood collected on

dried blood spot cards). The third (saliva samples), were not stored in a preservative and, for this reason, will have a distinct profile compared to other timepoints. This limits the context in which these samples will be useful, but will still provide biological insights in more targeted biological studies.”

Did you notice any degree of visual hemolysis in the samples or measure hemolysis? Hemolysis was not specifically measured, but there were no visible signs during collection. Samples were processed in a consistent timeframe and ordering post-collection to standardize any potential variance from sample degradation post-collection to cryo-preservation.

The authors mention that standardization is critical for the later utilization and interpretation of the samples. I agree. Which standards were followed (CLIA, ISBER, other)?

We follow ISBER standards. These integrated recommendations regarding biospecimen collection/processing and biospecimen storage include:

- We avoid cross-contamination of samples by keeping them separated during transportation and switching pipette and serological tips between all specimen fluid transfers.
- We use cryopreservation to transport samples or validated room temperature nucleic acid stabilizers (DNA/RNA Shield).
- We aliquot biospecimens on-site, to avoid introducing unnecessary freeze-thaw cycles.
- Samples that required cryopreservation were shipped on dry ice at the fastest shipping speed.
- Plasma and serum were stored at -80C. PBMCs were frozen with DMSO and a slow freezing process (Mr. Frosty) to avoid intracellular damage.
- Each sample/aliquot is assigned a unique barcode. These are either printed on the specimen container or on the sides of the containers (as opposed to the caps, which can be easily mixed). Our primary barcodes include:

Tube Type	Link
ThermoFisher 2D Barcoded Tubes	https://www.thermofisher.com/order/catalog/product/3742AMB?SID=srch-srp-3742AMB
Corning Cryovials with 1D Barcodes	https://ecatalog.corning.com/life-sciences/b2c/US/en/General-Labware/Cryogenic-Storage-and-Accessories/Cryogenic-Vials/Corning%C2%AE-Bar-Coded-External-and-Internal-Threaded-Polypropylene-Cryogenic-Vials/p/8672
Cryogenic SimPeel Labels	https://www.labtag.com/shop/product/dymo-compatible-cryogenic-labels-with-simpeel-technology-patent-pending-1-x-1-

I like the overview in Table 1. But Table 1 is massive and could preferably be divided into smaller focused samples based on the sample type. You can also try to reduce the white space and adjust the fonts and line spacing as simple measures. The duration is not only listed in days, but also years and months; however, for comparison across studies, it would help to have at least the duration in days for all studies. Somewhere in the table you have whole blood and serum in column one, and hematology in column 2, but hematology can only be measured on whole blood. In some of the rows for column 3 you have mentioned locations instead of participant N. It is not clear if each line represent one space mission, or the same space mission can cover several lines; therefore it would be great to add the name of mission and the year of the mission. Sometimes you mention sex and sometimes not (please be consistent or write if data on this is missing). Sometimes in column 3, you write “astronauts”, but aren’t all these studies for astronauts?

Thank you for the thorough review of the table. We’ve made the suggested changes, including breaking up the studies by sample type, standardized durations in terms of days, and clarified inconsistencies in terminology.

We are unable to consistently report the mission in the table as missions are not consistently reported in the literature we reference. We suspect this is due to identification of the astronaut participants and was likely a limitation placed on researcher’s accessing the data. NASA removes any metadata that could be used to identify astronaut genomic research participants. Unfortunately it is not possible as a reader to know which papers refer to the same astronauts.

For all tables provide a legend describing ALL the abbreviations.

Thank you for catching this. We’ve added a legend for abbreviations to the table descriptions.

It is not clear from table 2, column 3, which measurements are already measured on fresh samples, which were measured on frozen samples, and which are intended to be measured in the future on biobanked samples. You write “biobanking” for some but not all.

We added Extended Data Figure 1 to help clarify which assays were performed on fresh samples. (FYI in the revised manuscript, table 2 is now table 3). We’ve specified in the table description that “samples collected in excess were biobanked to enable additional experiments as new assays are developed.”

Extended Data Figure 1:

Table 3 should also include genomic DNA and RNA I guess.

Table 3 (now table 4 in the revised manuscript) is describing how samples were aliquoted before transfer to Weill Cornell Medicine. We've added columns to specify storage conditions at the field laboratories vs the central lab at WCM to clarify this. Extended Data Fig 1 has been included to help clarify exactly what was performed in field laboratory locations.

Describe the geographical locations of the processing and the biobanking. And was it centralized or decentralized?

We've added context throughout the paper to clarify what was performed at field laboratories and what was performed back at our central lab at WCM. Specifically, we've added some clarification to the abstract and the "Biospecimen Collection Overview" sections.

Describe time from sampling to processing and freezing for all samples. Were some sample types frozen before processing? What was the temperature of the samples pre-processing (ambient temperature, refrigerator, or frozen) after collection? When did the aliquoting take place (duration from the primary collection and what was the temperature of samples between collection and aliquoting)?

The following points of clarification were added to the "sample collection overview" and "blood tube and derivative" sections:

- All samples from ground timepoints were processed within 16 hours of collection.
- All samples from flight were processed immediately after retrieval from flight.
- Samples were stored at -80°C at each site immediately after processing and aliquoting.
- All samples were shipped via FedEx using Overnight Shipping to Weill Cornell Medicine for storage in the Cornell Aerospace Medicine Biobank (CAMbank) within 1 week of collection on dry ice in styrofoam boxes and spent <2 days in transit.
- No samples were shipped on a Thursday or Friday to avoid weekend shipping delays.
- All samples arrived at Weill Cornell Medicine with dry-ice still in the packaging and samples frozen.
- Samples were immediately transferred to -80°C for long term storage and biobanking.

- Tubes were transported from the collection site to the field laboratory at room temperature.

Some samples have a limited time window from primary tube collection to processing to ensure the stability of analytes. Please add the time windows for the samples.

We agree this is an important aspect to address. This has been addressed in our response to the previous comment, where we clarified various aspects of the sampling timeline.

Add the specific color of the “stopper” (the exact color is provided by the company) and colors also indicate type of material.

This has been added to a column in Table 3 (previously Table 2).

Write out BD for the vacutainer type.

This was added to Table 3 (previously Table 2) and to the first time each tube is mentioned in the text.

Which grade of DBS did you use?

We used Whatman 903 Protein Saver DBS. This was specified in the methods section, but has also now been moved to the main text.

Who among the authors was responsible for the standard operating procedures and for ensuring that the collection, processing, aliquoting, and storage were performed according to the SOPs (quality assurance)?

Three co-authors were responsible for this and had full awareness and training of protocols: Eliah Overbey, Krista Ryon, and Chris Mason.

Was any of the authors or associated personnel trained in biobanking procedures (there are various certificates and courses in this, some endorsed by ISBER).

Throughout the study we were using ISBER protocols for guidance. We acknowledge that it is important to receive formal training and integrate into the biobanking community to ensure best practices are continually followed, which we did for this study whenever possible. Even though we were already aligned with the ISBER methods,, Dr. Overbey has now formally completed the ISBER Biobank Education modules, and others in the lab are also now getting training. A screenshot of the certificate of completion is provided below:

Editorial Note: Image below redacted where no permission to publish could be obtained.

[REDACTED]

Have you planned how to prospectively control/check the quality of the samples? Some studies set aside certain aliquots for this, and at regular intervals check for some biomarkers, so a degradation profile can be estimated.

The details of how our biobank is being administered warrants its own paper and is in our pipeline as we add samples from Ax-2 and Polaris Dawn missions. Our plan for sample integrity is to continuously sample from a ground control cohort at regular intervals across multiple missions. The degradation of these samples can be analyzed as a proxy for the spaceflight samples. This will allow us to assess likely degradation of spaceflight samples without having to deplete limited astronaut samples across time.

It is not quite clear if any biomarkers were measured on fresh samples. If so, which?

The single-cell assays (10X Genomics Multiome Kit and Immune Repertoire Profiling) and the Quest Diagnostics tests (Comprehensive Metabolic Panel and Complete Blood Count) were performed on fresh samples. Extended Data Fig 1 (reported above) should help clarify this.

Be aware that some of the biomarkers you mention (such as cortisol and cytokines) have extremely strict requirements for collection (time of day for cortisol, time on the bench for cytokines) and are sensitive to long-term storage (cytokines).

We are aware of the limitations of these assays. Some tests, such as cortisol testing, may not have enough samples for a statistically significant study until we have more samples in the biobank due to restrictions that were placed on us surrounding time of collection that was beyond our control. For degradation-related issues, we have begun collecting samples from ground control participants that can be used to benchmark degradation without having to expend astronaut samples. We are also working with NASA to enable sample sharing to boost sample sizes by analyzing samples across multiple missions.

I am not sure exactly where the biobank is. Can you clarify?

The biobank (CAMbank) is located at Weill Cornell Medicine in New York City. We have included this location in the “biospecimen collection overview” section.

All abbreviations should be mentioned at least once in full writing.

We’ve modified various text sections to clarify various abbreviations. (L-, FD, R+, cfDNA, PBMCs, CPT, SST, etc).

Were the lab(s) CLIA certified?

The Comprehensive Metabolic Panel and Complete Blood Count assays were performed at Quest Diagnostics, which is CLIA certified, and one aliquot for whole genome sequencing will be performed in a CLIA lab. The remaining assays are research-use assays.

You use the term “established the standard” in the title. What exactly do you mean by that? Standardization can mean many things in the clinical laboratory: harmonization (that procedures are the same), but it can also refer to reference measurement procedures and reference materials. Usually, it is governmental agencies or larger organizations that set the standards for the clinical laboratory. Thus, please clarify what you mean by standardization in the title and in the manuscript.

We agree it's essential that this is clear to readers. We have included in the introduction the following statements:

- A key goal of SOMA is to standardize biospecimen collection and processing for spaceflight, to generate high-quality multi-omics data across spaceflight investigations, and to enable follow-up experiments with viably frozen cells and biobanked samples.
- This paper provides sample collection methods built for standardized field collections across different crews and missions.
- These protocols can generate harmonized datasets with greater statistical power and thus increase our scientific return yields from spaceflight investigations.

What time of the day were samples taken? And fasting/non-fasting considerations?

There were no fasting/non-fasting considerations. We asked for morning collections for all collection time points from SpaceX, which was followed whenever possible. Collection of

samples from commercial crews differs from NASA in that we do not have control over the crew's schedule and are given very limited sample collection windows.

Reviewer #2 (Remarks to the Author):

The study by Overbey and co-workers has a specific focus on collection and banking of biospecimens within an exceptional context, which is the SpaceX Inspiration4 mission, with the aim of studying the impact of spaceflight on the human body. The novelty of the study certainly resides in the exceptional source of these samples, however the impact these collections may have on further studies is less clear.

These methods are relevant to a handful of papers that are under review at the Nature Publishing Group. Here is an overview of the papers covering the Inspiration4 mission that have been submitted (this paper is in yellow):

Furthermore, we are employing and building off of these methods for our sample collection endeavors in the Ax-2 and Polaris Dawn missions. We've already collected blood and urine from the completed Ax-2 mission and pre-flight ground timepoints from the upcoming Polaris Dawn mission.

The collection is well described in itself, even with a pleonastic amount of details.

On the other side the study is mainly based on a description of standard parameters related to quality of samples and their analytes (testing of DNA, RNA, proteins, metabolites, and other biomolecules).

This is a correct description. This paper's intent is to provide a complete sample collection methodology description for the above papers. Additionally, we intend for this paper to be a resource to others who have the opportunity to collect astronaut biospecimens and would like to standardize their collection to existing datasets to boost their analyses potential. For example, the Translational Research Institute of Space Health at NASA has already started using the protocols from this paper.

Not all of the samples were collected during the flight. Histological samples in the form of skin biopsies were collected at ground timepoints. It is unclear how skin biopsies can be informative and exploited for instance for spatial genomic analyses (such as spatially resolved, single-molecule, and single-cell assays).

To understand the insights we can gain from spatially resolved transcriptomics of skin tissue, readers should refer to the Park *et al.* paper, "Spatial multi-omics of human skin reveals KRAS and inflammatory responses to spaceflight." This also included shotgun DNA and shotgun RNA sequencing for an analysis of the microbiome relative to the skin data. However, we have worked to clarify the scope of this study as a description of field sample collection and stabilization before the samples reached our central lab at Weill Cornell Medicine, which hopefully clarifies questions about sample suitability for these assays.

Although the effort in the meticulous collection should be praised, the study is very descriptive in nature and the data reported do not provide a significant impact on our knowledge.

This is the first flight of this new protocol, which is the most in-depth characterization of astronauts to date, and we note the significance for the protocol is anchored in three ways: (1) these same methods are now being deployed by the NASA Omics Archive (NAO), (2) these protocols are also being used by the Translational Research Institute for Space Health (TRISH) at Baylor College of Medicine / NASA, and (3) our protocols subsume and expand beyond the metrics of any space agency, and are being currently used for upcoming missions with Axiom Space, SpaceX, NASA, and ESA. Over time, these well-detailed protocols will lead to greater and more impactful comparisons between cohorts and data sets, and the ability to compare between missions will be invaluable for a better understanding of the cellular and molecular responses to spaceflight.

Moreover, the transparency of our protocols can help others collect samples with the same methodology, which itself can help the persistent issue of "low n" missions, enabling a means to combine and compare data across missions, which will be essential to separating true signal from spaceflight from any noise in the data.

Finally, we note the several IRBs that enabled this work, which are now included: Weill Cornell Medicine IRB# 21-05023569, WCG-IRB 1309934, "Multimodal Evaluation of Spaceflight Participants Health (MESH) - SpaceX Inspiration-4 Mission;" Translational Research Institute for Space Health IRB# 1316696, WCG-IRB 20214456, "Commercial Spaceflight Data Repository (CADRE)."

It is also questionable whether there will be a systematic collection of histological samples from personnel for these flights that will be used for research studies. We cannot exclude this will be the case, and in this respect this precise indication on best practice of collection of samples will be useful, yet not ground-breaking.

To our knowledge, there has been no histological staining on astronaut biopsies prior to this study. It is true each crew member needs to be consented, and not all will approve every assay requested. Also of note, our protocol for tissue collection has enabled the collection of moles after removal for one of the consented crew members, which also are undergoing histology and spatial omics analysis. Since tissue collections are opportunistic, based on the mission and crew engagement, we will utilize them whenever possible, and so far, all crews have signed our consent forms for this engagement.

We can foresee that the most relevant biospecimens that will be systematically collected from these units of personnel will be blood and urine to monitor health status of individuals involved in these missions. Again, the paper offers suggestions for good practice in this respect without leading to major advances.

Since the Twins Study in 2019 was the first in-depth omics profile of any astronauts, our goal for some of these more standard assays was indeed to keep them as consistent as possible. The value of our updated paper and methods is that the tubes, methods, and processing protocols replicate prior work from 2019 (e.g. CPT tubes), but also enable a broader range of assays and measures from the i4 crew and other crews (e.g. Polaris Dawn and Axiom-2, and Streck and SST tubes). Moreover, the Twins Study featured only about 260 samples from two crew members, whereas in this paper, we collected >2,911 samples from the crew members and the spacecraft, so our scale is an order of magnitude greater in terms of density of measurements.

REVIEWERS' COMMENTS

Reviewer #1 (Remarks to the Author):

The authors have responded adequately in the response and the revised manuscript.

We were happy to see we have addressed all reviewer concerns. We have included our response to the final comment in blue below.

REVIEWERS' COMMENTS

Reviewer #1 (Remarks to the Author):

The authors have responded adequately in the response and the revised manuscript.

We are glad to hear we have made adequate updates. Thank you for your time and help in the revisions of our manuscript.